# Influence of Thermal Modification in Nitrogen Atmosphere on the Selected Mechanical Properties of Black Poplar Wood (*Populus nigra* L.)

**DOI:** 10.3390/ma15227949

**Published:** 2022-11-10

**Authors:** Olga Bytner, Michał Drożdżek, Agnieszka Laskowska, Janusz Zawadzki

**Affiliations:** The Institute of Wood Sciences and Furniture, 159 Nowoursynowska St., 02-776 Warsaw, Poland

**Keywords:** bending strength, compressive strength, modulus of elasticity, thermal modification, nitrogen atmosphere, poplar wood

## Abstract

The thermal modification of wood in a nitrogen atmosphere shapes the functional properties of wood. The aim of this research was to determine the influence of different levels of temperature and the duration of thermal modification on the mechanical properties of the black poplar (*Populus nigra* L.). Black poplar was thermally modified in nitrogen atmosphere in the temperature range from 160 °C to 220 °C (6 levels) for 2 h to 8 h (4 levels), resulting in a total of 24 treatments. The effect of these treatments on compressive strength parallel to the grain (CS), modulus of rupture (MOR), and modulus of elasticity during bending (MOE) were analyzed. Thermal modification influenced the mechanical properties of black poplar wood. After thermal modification occurred in a nitrogen atmosphere, an increase in compressive strength was noticeable for all variants of black poplar wood modification. The highest 16% increase in CS was obtained for the modification carried out at the temperature of 160 °C and for 2 h. An increase was also found for MOE when modified under mild conditions, while a decrease occurred for variants at higher temperatures, i.e., for 200 °C and 220 °C. The study showed that for all modification variants, there was a decrease in MOR alongside the increase in modification temperature and time.

## 1. Introduction

Poplar is a fast-growing species that produces gentle colored wood with low density. It is one of the most efficient trees in terms of sustainability, having one of the highest growth rates and being the most inexpensive wood in the world. In Europe, one cubic meter of poplar can be produced, on average, in 15 years. On the other hand, oak takes 100 years [1]. Black poplar plays an important role in the wood industry. The advantage of black poplar consists in its plantation potential, resulting from a fast increase of biomass [2]. Currently, black poplar wood is mainly used as a raw material source in the cellulose industry or as a fuel. Therefore, poplar wood modification enables wide range of its innovative applications and, consequently, shapes its new position on the market of wood and wood-based materials.

Wood after heat treatment can be used in many applications. Among them are interior applications, i.e., for flooring, wall panels, and furniture. Thermally modified wood is also popular in exterior application, i.e., for elevation, garden furniture, and terraces. The main problem of the wood is its degradation by weathering. There are many factors that affect wood such as rain, snow, wind, temperature, and UV radiation [3]. Wood after heat treatment is more resistance to these conditions. This kind of wood has small changes in discoloration and surface cracks compared to native wood, as heat treatment reduces the wettability of wood [4,5,6].

Thermal modification is the action of applying high temperature to wood, usually in the range from 160 °C to 230 °C [7,8]. Currently, thermal modification is being studied, in particular, due to the growing demand for the development of methods to improve the durability of wood, while avoiding the use of toxic substances. Thermal modification meets these needs because it is an ecological process [9,10]. Currently, the most widely used wood modifications of high commercial importance are ThermoWood^®^ of Finland, Retification^®^ and Torrefaction^®^ of France, PLATO-wood^®^ of the Netherlands, and Oil-Heat Treatment (OHT) of Germany [11,12]. The main difference between these modifications is the type of atmosphere (nitrogen, water vapor, oil). One of the currently implemented commercially available methods is the modification of wood in a nitrogen atmosphere. This process, initially developed in France by Company NOW (New Option Wood), corresponds to a gentle pyrolysis of wood in an inert atmosphere [13]. Wood that can be modified in the nitrogen atmosphere include, among others, pine, spruce, birch, poplar, oak, ash, and locust wood [14,15].

The chemical, physical, and mechanical properties of wood subjected to heat treatment change permanently [16]. The scope of these changes depends on the type of treatment, its duration, and the temperature used, as well as the species of wood used [17]. Heating at high temperatures causes changes in the chemical structure of lignocellulosic materials [18]. The modification causes irreversible changes in the structure of the material. The modification temperature significantly affects the content of structural compounds (except for lignin) and extractives [19]. Hemicelluloses are the most sensitive, which are changed into new compounds such as formic and acetic acids, among others, resulting in a reduction of free hydroxyl groups in their chemical structures. Lignin is the most resistant structural compound in wood regarding temperature. During heat treatment it depolymerizes only cellulose with a low degree of polymerization and cellulose from an amorphous area. In this way, it grows up the index of crystallinity cellulose [20,21,22,23]. These changes in chemical composition influence the physical and mechanical properties of wood. The smaller number of hydrogen bonds result in lower water absorption. The material has lower equilibrium moisture content [24], and lower swelling parameters [25].

Regardless of the wood species used and the process, similar relationships are observed. Thermal modification causes a decrease in the mass and density of wood, a change of color to a darker one, a decrease of shrinking and swelling due to reduced equilibrium moisture content of the wood [19,26,27], and changing mechanical properties [28,29]. Overall, it can be concluded that there is only a slight change in the modulus of elasticity (MOE), but the decrease in modulus of rupture (MOR) is significant regardless of the process or wood species [30]. The study conducted by Kozakiewicz et al. [31] showed that wood properties such as compressive strength parallel to the grain (CS), modulus of elasticity during bending (MOE), and compressing (MCS) did not differ significantly depending on the temperature of the modification process in superheated steam. It was found that these properties did not depend, to a large extent, on changes in the chemical composition of wood. In the literature, in some cases, an increase of MOE was observed after thermal modification [32,33]. The higher the temperature of the modification process, the more important the reduction of MOR and Brinell hardness (BH) is [31,33]. Yildiz et al. [34] noted that the MOR of thermally modified wood may drop from 10% to 30%, while Bekhta and Niemz [35], Candelier et al. [36], Sandberg and Kutnar [30] described a reduction of wood MOR by up to 50%. After wood is subjected to thermal treatment, its brittleness increases, which significantly reduces MOR [30].

Lower mechanical properties can significantly limit the use of products after modification. Therefore, it is important to optimize processes by the appropriate selection of parameters, where the time and temperature of the process are considered the most important factors. In addition, the type of atmosphere used may significantly affect the degree of degradation processes during modification. The thermal degradation of wood heated in the oxygen atmosphere is faster than that of wood heated in an oxygen-free atmosphere. The decrease in mechanical properties can be reduced by using a closed system with an inert gas, i.e., nitrogen. Thermal modification in nitrogen is less destructive compared to the modification in a water vapor atmosphere. It was stated that the mass loss of black poplar modified in superheated steam for 2 h at 160 °C, 190 °C, and 220 °C was 3%, 4%, and 12%, respectively, while for the modification in nitrogen these values were 1%, 1%, and 7% [19,31]. The aim of the study was to determine the influence of technological parameters of modification on the mechanical properties of black poplar (*Populus nigra* L.) wood. It should be expected that the research will broaden the knowledge on the modification of low-density wood species, such as black poplar, in the context of the possibility of its use as a substitute replacing exotic wood.

## 2. Materials and Methods

### 2.1. Black Poplar Wood Samples

In this case, 40-year-old black poplar (*Populus nigra* L.) came from a forest in Poland, in the Eastern part of the Mazovian province, State Forest District Sokołów Podlaski. The trees had a diameter at breast height (DBH) up to 0.5 m and a mean growth ring width greater than 5 mm. Boards were conditioned in a normal climate (temperature 20 °C ± 2 °C, relative humidity 65% ± 5%) to an air-dry condition. About 1500 samples of 20 mm × 20 mm × 300 mm (radial × tangential × longitudinal) were prepared. The highest quality samples were obtained from the wood of the black poplar, with no visible material defects (knots, slop of grain, resin pocked, bark pocked, reaction wood, wanes, blue stains, decays, insect holes, shakes, distortions, etc.). The surfaces of the wood samples were finished by planning. The samples were sorted by density. Each group of 30 samples have similar average density and standard deviation, i.e., 391 ± 39 kg × m^−3^. The moisture content of poplar wood was measured according to ISO 13061-1:2014 [37] and density was determined according to the ISO 13061-2:2014 standard [38]. One group of samples was the control group (native wood) and the other consisted of those thermally modified in nitrogen atmosphere.

### 2.2. Black Poplar Wood Thermal Modification in Nitrogen

The process of thermal modification of black poplar wood was carried out in a nitrogen atmosphere according to the procedure given by Bytner et al. [19]. The modification of black poplar wood was carried out at the temperatures of 160 °C, 170 °C, 180 °C, 190 °C, 200 °C, and 220 °C. The modification times (for each of the modification temperatures) were 2 h, 4 h, 6 h, and 8 h. Each variant of modification was performed on 30 samples. The modification process was carried out in an 0.25 m^3^ chamber (Explo Solutions sp. z o. o., Warsaw, Poland), with a temperature control system and air gas circulation.

### 2.3. Determination of the Properties of Thermally Modified Wood

The mass loss (*ML*) after modification was expressed as a percentage of the initial mass of the totally dry wood. The *ML* was calculated according to Equation (1), where *m_o_* is the mass of the oven-dried wood (g), and *m_m_* is the mass of the oven-dried wood after thermal modification (g):(1)$$ML=\frac{m_{o}-m_{m}}{m_{o}}\times 100\;\left(\%\right),$$

After the modification process, samples were dried in a dryer at 103 °C for 12 h. The compressive strength parallel to grain (CS) was determined in accordance with ISO 13061-17:2017 [39]. Wood blocks with dimensions of 20 mm × 20 mm × 30 mm (radial × tangential × longitudinal) were cut from each 30 treated samples for testing CS of the black poplar. The tests of CS were carried out by using a computer program coupled with Instron^®^ testing machine, model 3382 (Norwood, MA, USA).

The modulus of rupture (MOR) and modulus of elasticity (MOE) tests were conducted in accordance with the ISO 13061-3:2014 [40] and ISO 13061-4:2014 [41] standards, respectively. In this case, 30 samples with dimensions of 20 mm × 20 mm × 300 mm (radial × tangential × longitudinal) were used for testing the MOR and MOE of the black poplar. The MOR and MOE research was carried out using a computer program coupled with an Instron^®^ testing machine, model 3369 (Instron^®^, Norwood, MA, USA). The wood properties were determined for 30 control samples and for each variant of thermal modification.

### 2.4. Statistical Analysis

The STATISTICA Version-13.3 software of StatSoft, Inc. (TIBCO Software Inc., Palo Alto, CA, USA) was used to carry out the statistical analyses (*t*-test, ANOVA, Fischer’s F-test). The statistical analyses were performed at the significance level (*p*) of 0.05. The percentual impact of the temperature and time of modification, the so-called Factor Influence, on the physical and mechanical properties of black poplar wood that was thermally modified in a nitrogen atmosphere was presented. The control group was native (untreated) wood.

## 3. Results and Discussion

As a result of thermal modification, the density of black poplar wood decreased (Table 1), which was a consequence of the mass loss caused by the degradation of the structural components of wood, mainly hemicelluloses [19]. The mass loss (ML) results are presented in Table 1, which ranged between 0.7% ± 0.4% and 14.0% ± 1.1% and increased with increased modification temperature and time. A similar trend for ML was also reported by Cademartori [42] for fast-growing Gympie messmate wood subjected to two-step steam-heat treatments. Mass loss at temperatures between 160 °C and 180 °C is caused by evaporation of water (bound and free water), while at temperatures from 180 °C to 220 °C ML it is the effect of the decomposition of the least stable hemicelluloses [43,44].

The density of native black poplar wood with 0% moisture content was 391 ± 39 kg × m^−3^. It is a typical density of poplar wood obtained at a young harvest age (approx. 40 years), characterized by a large share of juvenile wood in the research material—wood of naturally lower density compared to mature wood. The density of the tested black poplar wood is consistent with the literature data. According to Wagenführ [46] the density of black poplar wood with a moisture content of 0% ranges from 370 kg × m^−3^ to 520 kg × m^−3^. A significant decrease in density of thermally modified black poplar, as compared to native wood, was observed at higher temperatures (Table 4), the maximum (13% decrease) of which was reached for extreme conditions, i.e., 220 °C and for 8 h, where the density was 340 ± 38 kg × m^−3^ (Table 2).

The values of compressive strength parallel to the grain (CS) of black poplar wood modified in nitrogen atmosphere were presented in Table 3. The native black poplar in an absolutely dry state was characterized by compressive strength equal to 48.9 ± 7.7 MPa. Wagenführ [46] stated that the CS of black poplar wood ranges from 26 MPa to 56 MPa. After thermal modification in a nitrogen atmosphere, the increase in compressive strength was noticeable for all variants of black poplar wood modification. It should be stated that, depending on the modification conditions, an increase in the CS value from approx. 1% to approx. 16% can be obtained. The highest increase in CS was obtained for the modification carried out at the temperature of 160 °C for 2 h, and the lowest for the temperature of 220 °C for 8 h. It follows that the higher modification temperature and the longer process time results in an increase in the CS value compared to the CS of native wood. It should be noted that a statistically significant increase in the CS value (Table 4) was recorded for black poplar wood modified at the temperature of 160 °C, 170 °C, 180 °C for 2 h, 4 h, 6 h (an increase in the range of 8–16%) and at the temperature of 190 °C, 200 °C for 2 h (9% increase).

The research shows that thermal modification has a positive effect on the compressive strength of wood. Percin et al. [47] obtained similar relationships. For beech (*Fagus orientalis* Lipsky) wood thermally modified under atmospheric pressure in almost all modification variants (for temperatures of 150 °C, 175 °C, 200 °C and time of 1 h, 3 h, 5 h), the authors obtained an increase in the value of compressive strength contained between 2.98% and 19.3%. A slight decrease in compressive strength at the level of 1.16% was shown only in the case of beech wood modified at 200 °C for 5 h. Kaymakci and Bayram [48] obtained an increase in the CS for poplar wood (*Populus alba* L.) after modification at the temperatures of 120 °C, 150 °C, 180 °C for 2 h and 4 h under atmospheric pressure. The highest (i.e., 14%) increase in the CS value was obtained after the modification of poplar wood at a temperature of 120 °C for 2 h. However, the decrease in the CS value was recorded only for the highest modification temperature, i.e., 210 °C and the times of 2 h and 4 h, with the maximum decrease in CS at the level of 22% for 210 °C and 4 h. The decrease in compressive strength can be reduced by using a closed system with an inert gas such as nitrogen or steam [34]. Moreover, the thermal degradation of wood proceeds faster in methods of modification with the presence of oxygen compared to modification carried out in an anaerobic environment. The increase of the compressive strength in a longitudinal direction might be due to a lower amount of bound water in heat treated wood [32]. Additionally, the reason may be the growth of crystalline cellulose through the degradation and crystallization of its amorphous areas and changes in the structure of lignin [32].

The modulus of rupture (MOR) results of unmodified (native) and modified black poplar wood in a nitrogen atmosphere were presented in Table 5. The MOR for untreated black poplar wood was 81.5 ± 15.0 MPa. Wagenführ [46] stated that the MOR of black poplar wood ranged from 43 MPa to 94 MPa. In general, it can be concluded that in all variants, there was a decrease in MOR with increasing modification temperature and time. The exception was the black poplar wood modified in a temperature of 160 °C for 2 h and 4 h, and a temperature of 170 °C modified for 2 h, for which no statistically significant changes in MOR were found compared to native wood (Table 7). The greatest decrease in bending strength, at the level of approx. 44%, was observed after modification at 220 °C for 8 h.

Bal [49] noted that thermally modified pine wood in a nitrogen atmosphere indicated a decrease in the MOR value, which was 2.7%, 1.5%, 19.8% for the modification temperatures of 180 °C, 200 °C, 220 °C and the modification time of 2.5 h. For the tested black poplar wood, the MOR for analogous temperatures and 2 h treatment time were, respectively, lower by 12%, 27% and 37%. The lower impact of thermal modification on MOR in the studies by Bal [49] may result from the softwood species used for the modification. Pine has a lower hemicelluloses content compared to hardwood [50]. During modification, hemicelluloses are mainly degraded, where the percentage content of this wood component is lower for softwood compared to hardwood, which may result in smaller percentage drops of MOR. In studies by Kaymakci and Bayram [48] for poplar wood (*Populus alba* L.) modified at temperatures of 120 °C, 150 °C, 180 °C, 210 °C for 2 h and 4 h under atmospheric pressure, a decrease in MOR from 3.67% to 46.25% was noted. The greatest decrease in the MOR value was obtained for the modification at 210 °C for 4 h. A significant decrease in the MOR value to the level of 46.25%, which is higher than for the tested black poplar, may be the result of the environment in which the thermal modification was carried out.

The decrease in MOR is mainly a result of the degradation of hemicelluloses during thermal modification. Hemicelluloses are depolymerized by hydrolysis reactions to oligomers and monomers. This involves cleavage of the side-chain constituents, i.e., arabinose and galactose, followed by the degradation of the main-chain constituents, i.e., mannose, glucose and xylose. The corresponding pentoses and hexoses are dehydrated to, respectively, furfural and hydroxymethylfurfural [32]. The decrease in MOR may also be affected by the degradation of cellulose chains with a low degree of polymerization, which is mostly amorphic cellulose [51]. Zawadzki et al. [52] indicated that for Scots pine thermally modified in superheated steam at the temperature of 200 °C, important changes were already observed in the molar mass of cellulose, as well as its depolymerization.

The values of the modulus of elasticity (MOE) of unmodified (native) and modified black popular wood in a nitrogen atmosphere were presented in Table 6. The MOE for untreated black poplar wood was 6435 ± 1100 MPa. Wagenführ [46] indicated that the MOE of black poplar wood ranged from 4000 MPa to 11,700 MPa.

Under the influence of the modification process, changes in the MOE value depending on the temperature and the modification time were noted. Only for higher temperatures and longer modification time (i.e., for modifications carried out at 220 °C for 2 h, 4 h, 6 h, 8 h and at 200 °C for 6 h and 8 h), a slight decrease in MOE was observed. For the remaining variants, the MOE values were higher or at the same level. It should be noted that statistically significant changes in MOE compared to native wood were noted for black poplar wood modified at 160 °C for 2 h and 6 h (Table 7). For the modification carried out at the temperature of 160 °C for 6 h, the highest increase of MOE was achieved, i.e., by 9%. The greatest decrease in MOE, i.e., by approx. 2%, was recorded for black poplar wood modified at 220 °C for 6 h and 8 h.

In the studies by Araújo et al. [53] *Mimosa scabrella* (bracatinga) were heated in an oven under nitrogen, at 180 °C, 200 °C, and 220 °C for 1 h. The MOE for wood modified at 180 °C increased by approx. 1%; however, for modified wood at 200 °C and 220 °C, it decreased by approx. 1%; such changes in MOE can be considered insignificant. Boonstra et al. [32] used an industrial two-stage heat treatment method under relative mild conditions (<200 °C). For modified Scots pine (*Pinus sylvestris* L.) wood, an increase in the MOE by 10% was recorded, while for radiata pine (*Pinus radiata* D.), the MOE value increased by 13% after modification at 165 °C for 0.5 h. Some researchers reported that the MOR is more affected than the MOE by the heat treatment [16,27,49]. For mild treatment conditions, the MOE often increases, whereas it decreases for severe treatment conditions [54]. This increase is explained by an increase in the crystallinity of cellulose and a decrease in the equilibrium moisture content (EMC) of wood [16]. The increase in the MOE of the thermally modified wood may be the result of increased lignin cross-linking that makes the structure around the cellulose microfibrils and the middle lamella more rigid [32,55].

Technological parameters had a slight influence on the differentiation of the density (MD) and the examined mechanical properties of black poplar wood (Figure 1). The modification temperature showed a significant effect on all of the analyzed properties, with the greatest impact on the MOR of black poplar wood (factor influence at the level of 37%). In the case of the other properties, i.e., MD, CS and MOE, the impact was below 10%. On the other hand, the modification time was responsible for a small extent for the variability of the CS and the MOR of black poplar wood (factor influence at the level of 4%). The interaction between temperature and modification time did not significantly affect the analyzed properties of wood. The indicated relationships confirm those presented in Table 4 and Table 7. For most of the analyzed variants of thermal modification in nitrogen atmosphere, there are no statistically significant differences (ns) in the MD, CS and MOE of black poplar wood. High error values (above 60%) indicate that there are other factors that can significantly influence the properties of thermally modified wood, e.g., wood species and the type of atmosphere. Kozakiewicz et al. [31] states that the moisture content of the thermally modified wood (in superheated steam) is an important factor in determining the character of changes in wood’s mechanical properties.

## 4. Conclusions

As a result of thermal modification in a nitrogen atmosphere, the mechanical properties of black poplar wood changes. The characteristics of the changes in compressive strength parallel to the grain, modulus of rupture, and modulus of elasticity during bending were different. The greatest influence of modification was noted for the modulus of rupture of black poplar wood. The decrease in the modulus of rupture progressed with increasing temperature and time, with a greater influence of temperature than time observed. The only exceptions were black poplar wood modified at a temperature of 160 °C for 2 h and 4 h, and at the temperature of 170 °C modified for 2 h, for which no statistically significant changes in the modulus of rupture were found compared to native (untreated) wood. In general, it could be concluded that thermal modification in a nitrogen atmosphere has a positive effect on the compressive strength parallel to the grain and modulus of elasticity during the bending of black poplar. Higher temperatures of modification and longer heating times influence a small increase in value of compressive strength compared to native (untreated) wood. The highest 16% increase in the compressive strength was obtained for the poplar wood modified at a temperature of 160 °C for 2 h. Black poplar wood modified at 160 °C for 6 h had the highest (9%) increase of modulus of elasticity.

## Figures and Tables

**Figure 1 materials-15-07949-f001:**
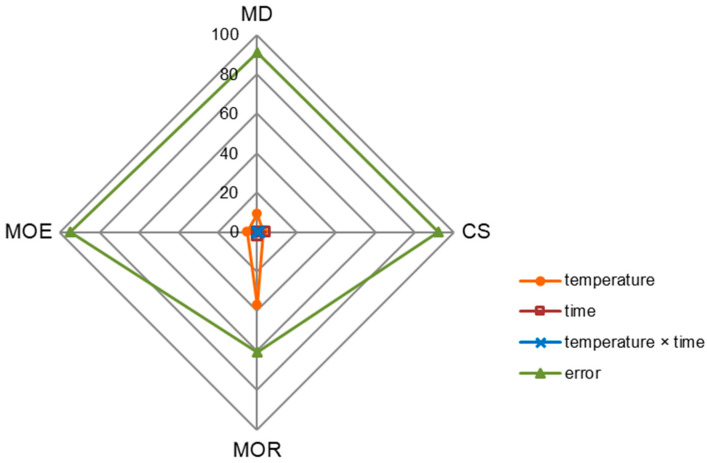
The percentual influence of temperature, modification time, interaction between temperature, and modification time on the density (MD), compressive strength (CS), modulus of rupture (MOR), and modulus of elasticity (MOE) of thermally modified black poplar.

**Table 1 materials-15-07949-t001:** The mass loss of black poplar wood thermally modified in nitrogen atmosphere; ±(SD), [45].

Modification Time (h)	Modification Temperature (°C)					
	160	170	180	190	200	220
	Mass Loss (%)					
2	0.9 ± 0.5	0.8 ± 0.1	0.8 ± 0.4	0.8 ± 0.4	4.0 ± 0.3	6.8 ± 0.9
4	0.7 ± 0.4	0.8 ± 0.1	1.9 ± 0.1	2.6 ± 0.1	5.4 ± 0.3	12.0 ± 0.9
6	0.7 ± 0.4	0.9 ± 0.5	2.2 ± 0.5	2.1 ± 0.7	5.8 ± 0.9	12.6 ± 0.9
8	0.9 ± 0.4	1.1 ± 0.1	3.0 ± 0.2	3.5 ± 0.6	7.1 ± 1.0	14.0 ± 1.1

**Table 2 materials-15-07949-t002:** The density of black poplar wood thermally modified in nitrogen atmosphere; ±(SD).

Modification Time (h)	Modification Temperature (°C)					
	160	170	180	190	200	220
	Density (kg × m^−3^)					
2	378 ± 37	376 ± 38	373 ± 38	373 ± 37	365 ± 35	346 ± 32
4	376 ± 38	371 ± 35	368 ± 36	369 ± 38	364 ± 40	345 ± 36
6	375 ± 30	371 ± 34	368 ± 32	365 ± 36	360 ± 37	342 ± 33
8	375 ± 39	371 ± 37	368 ± 38	364 ± 34	355 ± 34	340 ± 38

**Table 3 materials-15-07949-t003:** The compressive strength (CS) of black poplar wood thermally modified in nitrogen atmosphere; ±(SD).

**Native Poplar Wood**	48.9 ± 7.7 MPa					
**Modified Poplar Wood**	**Modification Temperature (°C)**					
	**160**	**170**	**180**	**190**	**200**	**220**
**Modification Time (h)**	**CS (MPa)**					
2	56.5 ± 7.5	56.2 ± 11.1	55.9 ± 8.4	53.4 ± 5.6	53.4 ± 5.0	51.2 ± 7.7
4	56.3 ± 9.1	55.9 ± 5.9	53.2 ± 6.4	52.9 ± 8.8	52.7 ± 9.1	51.0 ± 10.0
6	54.4 ± 6.3	54.2 ± 7.5	52.8 ± 5.8	52.5 ± 10.2	52.5 ± 9.3	50.8 ± 7.8
8	51.7 ± 3.3	50.3 ± 8.6	50.2 ± 3.2	50.2 ± 4.0	49.9 ± 4.4	49.4 ± 9.2

**Table 4 materials-15-07949-t004:** A statistical analysis of the density, compressive strength (CS) results (*t*-test, *p* ≤ 0.05, *—significant dependence, ns—no significant dependence).

Variants	Native Wood	160 °C_2 h	160 °C_4 h	160 °C_6 h	160 °C_8 h	170 °C_2 h	170 °C_4 h	170 °C_6 h	170 °C_8 h	180 °C_2 h	180 °C_4 h	180 °C_6 h	180 °C_8 h	190 °C_2 h	190 °C_4 h	190 °C_6 h	190 °C_8 h	200 °C_2 h	200 °C_4 h	200 °C_6 h	200 °C_8 h	220 °C_2 h	220 °C_4 h	220 °C_6 h	220 °C_8 h	
**Native Wood**		*	*	*	ns	*	*	*	ns	*	*	*	ns	*	ns	ns	ns	*	ns	ns	ns	ns	ns	ns	ns	**CS**
**160 °C_2 h**	ns		ns	ns	*	ns	ns	ns	*	ns	ns	*	*	ns	ns	ns	*	ns	ns	ns	*	*	*	*	*	
**160 °C_4 h**	ns	ns		ns	*	ns	ns	ns	*	ns	ns	ns	*	ns	ns	ns	*	ns	ns	ns	*	*	*	*	*	
**160 °C_6 h**	ns	ns	ns		*	ns	ns	ns	*	ns	ns	ns	*	ns	ns	ns	*	ns	ns	ns	*	ns	ns	ns	*	
**160 °C_8 h**	ns	ns	ns	ns		*	*	ns	ns	*	ns	ns	ns	ns	ns	ns	ns	ns	ns	ns	ns	ns	ns	ns	ns	
**170 °C_2 h**	ns	ns	ns	ns	ns		ns	ns	*	ns	ns	ns	*	ns	ns	ns	*	ns	ns	ns	*	*	ns	*	*	
**170 °C_4 h**	*	ns	ns	ns	ns	ns		ns	*	ns	ns	*	*	ns	ns	ns	*	ns	ns	ns	*	*	*	*	*	
**170 °C_6 h**	*	ns	ns	ns	ns	ns	ns		ns	ns	ns	ns	*	ns	ns	ns	*	ns	ns	ns	*	ns	ns	ns	*	
**170 °C_8 h**	ns	ns	ns	ns	ns	ns	ns	ns		*	ns	ns	ns	ns	ns	ns	ns	ns	ns	ns	ns	ns	ns	ns	ns	
**180 °C_2 h**	ns	ns	ns	ns	ns	ns	ns	ns	ns		ns	ns	*	ns	ns	ns	*	ns	ns	ns	*	*	*	*	*	
**180 °C_4 h**	*	ns	ns	ns	ns	ns	ns	ns	ns	ns		ns	*	ns	ns	ns	*	ns	ns	ns	*	ns	ns	ns	ns	
**180 °C_6 h**	*	ns	ns	ns	ns	ns	ns	ns	ns	ns	ns		*	ns	ns	ns	ns	ns	ns	ns	*	ns	ns	ns	ns	
**180 °C_8 h**	*	ns	ns	ns	ns	ns	ns	ns	ns	ns	ns	ns		*	ns	ns	ns	*	ns	ns	ns	ns	ns	ns	ns	
**190 °C_2 h**	ns	ns	ns	ns	ns	ns	ns	ns	ns	ns	ns	ns	ns		ns	ns	*	ns	ns	ns	*	ns	ns	ns	*	
**190 °C_4 h**	*	ns	ns	ns	ns	ns	ns	ns	ns	ns	ns	ns	ns	ns		ns	ns	ns	ns	ns	ns	ns	ns	ns	ns	
**190 °C_6 h**	*	ns	ns	ns	ns	ns	ns	ns	ns	ns	ns	ns	ns	ns	ns		ns	ns	ns	ns	ns	ns	ns	ns	ns	
**190 °C_8 h**	*	ns	ns	ns	ns	ns	ns	ns	ns	ns	ns	ns	ns	ns	ns	ns		*	ns	ns	ns	ns	ns	ns	ns	
**200 °C_2 h**	*	ns	ns	ns	ns	ns	ns	ns	ns	ns	ns	ns	ns	ns	ns	ns	ns		ns	ns	*	ns	ns	ns	*	
**200 °C_4 h**	*	ns	ns	ns	ns	ns	ns	ns	ns	ns	ns	ns	ns	ns	ns	ns	ns	ns		ns	ns	ns	ns	ns	ns	
**200 °C_6 h**	*	ns	ns	ns	ns	ns	ns	ns	ns	ns	ns	ns	ns	ns	ns	ns	ns	ns	ns		ns	ns	ns	ns	ns	
**200 °C_8 h**	*	*	*	*	*	*	ns	ns	ns	ns	ns	ns	ns	ns	ns	ns	ns	ns	ns	ns		ns	ns	ns	ns	
**220 °C_2 h**	*	*	*	*	*	*	*	*	*	*	*	*	*	*	*	*	*	*	ns	ns	ns		ns	ns	ns	
**220 °C_4 h**	*	*	*	*	*	*	*	*	*	*	*	*	*	*	*	*	*	*	ns	ns	ns	ns		ns	ns	
**220 °C_6 h**	*	*	*	*	*	*	*	*	*	*	*	*	*	*	*	*	*	*	*	*	ns	ns	ns		ns	
**220 °C_8 h**	*	*	*	*	*	*	*	*	*	*	*	*	*	*	*	*	*	*	*	*	ns	ns	ns	ns		
	**Density**																									

Colour in the table: yellow—statistical analysis results for the density, purple—statistical analysis results for the compressive strength (CS).

**Table 5 materials-15-07949-t005:** The modulus of rupture (MOR) of black poplar wood thermally modified in nitrogen atmosphere; ±(SD).

**Native Poplar Wood**	81.5 ± 15.0 MPa					
**Modified Poplar Wood**	**Modification Temperature (°C)**					
	**160**	**170**	**180**	**190**	**200**	**220**
**Modification Time (h)**	**MOR (MPa)**					
2	80.0 ± 13.8	74.5 ± 12.7	71.8 ± 13.6	70.8 ± 15.0	59.1 ± 12.9	51.2 ± 11.7
4	75.5 ± 14.3	73.8 ± 9.8	70.3 ± 15.1	65.4 ± 15.2	58.9 ± 8.3	47.1 ± 11.6
6	74.8 ± 10.0	73.6 ± 14.7	69.0 ± 13.4	64.8 ± 12.5	58.2 ± 13.5	47.0 ± 8.0
8	74.3 ± 11.0	72.8 ± 15.1	69.3 ± 13.9	61.8 ± 12.5	55.4 ± 13.1	45.5 ± 12.0

**Table 6 materials-15-07949-t006:** The modulus of elasticity (MOE) of black poplar wood thermally modified in nitrogen atmosphere; ±(SD).

**Native Poplar Wood**	6435 ± 1100 MPa					
**Modified Poplar Wood**	**Modification Temperature (°C)**					
	**160**	**170**	**180**	**190**	**200**	**220**
**Modification Time (h)**	**MOE (MPa)**					
2	6963 ± 641	6914 ± 1296	6889 ± 1099	6725 ± 856	6515 ± 824	6429 ± 522
4	6902 ± 1271	6822 ± 492	6623 ± 556	6540 ± 1404	6435 ± 842	6357 ± 1294
6	6997 ± 545	6830 ± 1184	6691 ± 1209	6524 ± 673	6328 ± 810	6305 ± 1275
8	6948 ± 972	6674 ± 742	6515 ± 997	6475 ± 809	6347 ± 1144	6327 ± 1400

**Table 7 materials-15-07949-t007:** A statistical analysis of the modulus of rupture (MOR), modulus of elasticity (MOE) results (*t*-test, *p* ≤ 0.05, *—significant dependence, ns—no significant dependence).

Variants	Native Wood	160 °C_2 h	160 °C_4 h	160 °C_6 h	160 °C_8 h	170 °C_2 h	170 °C_4 h	170 °C_6 h	170 °C_8 h	180 °C_2 h	180 °C_4 h	180 °C_6 h	180 °C_8 h	190 °C_2 h	190 °C_4 h	190 °C_6 h	190 °C_8 h	200 °C_2 h	200 °C_4 h	200 °C_6 h	200 °C_8 h	220 °C_2 h	220 °C_4 h	220 °C_6 h	220 °C_8 h	
**Native Wood**		*	ns	*	ns	ns	ns	ns	ns	ns	ns	ns	ns	ns	ns	ns	ns	ns	ns	ns	ns	ns	ns	ns	ns	**MOE**
**160 °C_2 h**	ns		ns	ns	ns	ns	ns	ns	ns	ns	*	ns	*	ns	ns	*	*	*	*	*	*	*	*	*	*	
**160 °C_4 h**	ns	ns		ns	ns	ns	ns	ns	ns	ns	ns	ns	ns	ns	ns	ns	ns	ns	ns	*	ns	ns	ns	ns	ns	
**160 °C_6 h**	*	ns	ns		ns	ns	ns	ns	ns	ns	*	ns	*	ns	ns	*	*	*	*	*	*	*	*	*	*	
**160 °C_8 h**	*	ns	ns	ns		ns	ns	ns	ns	ns	ns	ns	ns	ns	ns	ns	*	ns	*	*	*	*	ns	*	ns	
**170 °C_2 h**	ns	ns	ns	ns	ns		ns	ns	ns	ns	ns	ns	ns	ns	ns	ns	ns	ns	ns	*	ns	ns	ns	ns	ns	
**170 °C_4 h**	*	ns	ns	ns	ns	ns		ns	ns	ns	ns	ns	ns	ns	ns	ns	*	ns	*	*	*	*	ns	*	ns	
**170 °C_6 h**	*	ns	ns	ns	ns	ns	ns		ns	ns	ns	ns	ns	ns	ns	ns	ns	ns	ns	ns	ns	ns	ns	ns	ns	
**170 °C_8 h**	*	ns	ns	ns	ns	ns	ns	ns		ns	ns	ns	ns	ns	ns	ns	ns	ns	ns	ns	ns	ns	ns	ns	ns	
**180 °C_2 h**	*	*	ns	ns	ns	ns	ns	ns	ns		ns	ns	ns	ns	ns	ns	ns	ns	ns	*	ns	*	ns	ns	ns	
**180 °C_4 h**	*	*	ns	ns	ns	ns	ns	ns	ns	ns		ns	ns	ns	ns	ns	ns	ns	ns	ns	ns	ns	ns	ns	ns	
**180 °C_6 h**	*	*	ns	ns	ns	ns	ns	ns	ns	ns	ns		ns	ns	ns	ns	ns	ns	ns	ns	ns	ns	ns	ns	ns	
**180 °C_8 h**	*	*	ns	ns	ns	ns	ns	ns	ns	ns	ns	ns		ns	ns	ns	ns	ns	ns	ns	ns	ns	ns	ns	ns	
**190 °C_2 h**	*	*	ns	ns	ns	ns	ns	ns	ns	ns	ns	ns	ns		ns	ns	ns	ns	ns	ns	ns	ns	ns	ns	ns	
**190 °C_4 h**	*	*	*	*	*	*	*	*	ns	ns	ns	ns	ns	ns		ns	ns	ns	ns	ns	ns	ns	ns	ns	ns	
**190 °C_6 h**	*	*	*	*	*	*	*	*	*	*	ns	ns	ns	ns	ns		ns	ns	ns	ns	ns	ns	ns	ns	ns	
**190 °C_8 h**	*	*	*	*	*	*	*	*	*	*	*	*	*	*	ns	ns		ns	ns	ns	ns	ns	ns	ns	ns	
**200 °C_2 h**	*	*	*	*	*	*	*	*	*	*	*	*	*	*	ns	ns	ns		ns	ns	ns	ns	ns	ns	ns	
**200 °C_4 h**	*	*	*	*	*	*	*	*	*	*	*	*	*	*	*	*	ns	ns		ns	ns	ns	ns	ns	ns	
**200 °C_6 h**	*	*	*	*	*	*	*	*	*	*	*	*	*	*	ns	ns	ns	ns	ns		ns	ns	ns	ns	ns	
**200 °C_8 h**	*	*	*	*	*	*	*	*	*	*	*	*	*	*	*	*	ns	ns	ns	ns		ns	ns	ns	ns	
**220 °C_2 h**	*	*	*	*	*	*	*	*	*	*	*	*	*	*	*	*	*	*	*	*	ns		ns	ns	ns	
**220 °C_4 h**	*	*	*	*	*	*	*	*	*	*	*	*	*	*	*	*	*	*	*	*	*	ns		ns	ns	
**220 °C_6 h**	*	*	*	*	*	*	*	*	*	*	*	*	*	*	*	*	*	*	*	*	*	ns	ns		ns	
**220 °C_8 h**	*	*	*	*	*	*	*	*	*	*	*	*	*	*	*	*	*	*	*	*	*	ns	ns	ns		
	**MOR**																									

Colour in the table: yellow—statistical analysis results for the modulus of rupture (MOR), purple—statistical analysis results for the modulus of elasticity (MOE).

## Data Availability

The data presented in this study are available on request from the corresponding author.

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
