# Peer review of "Influence of Thermal Modification in Nitrogen Atmosphere on the Selected Mechanical Properties of Black Poplar Wood (Populus nigra L.)"

_materials, 2022, doi:10.3390/ma15227949_

Round 1
Reviewer 1 Report
Dear Authors,
Your manuscript addresses an interesting topic. I find that the result of this research is a useful addition to the literature and can contribute to the optimization of thermal treatment and valorization of black poplar wood.
Following is the outcome of my review to be implemented in the revised manuscript:
- I have minor corrections and suggestions in the text which you find in red in the enclosed PDF.
- The "introduction" of the paper doesn't really let the reader know about the applications for which black poplar heat-treated wood could be of interest, and the respective physical or mechanical properties required for such applications. Perhaps you could add more specific information (rather than general) into the introduction.
- The "conclusions" don’t do justice to your work. I would like you to rewrite them with a focus on the following points rather than rephrasing the results:
· The extent to which the thermal modification of black poplar using the experimental factors in your study could achieve the objective of expanding the existing knowledge in the field. Please mention the impact of thermal treatment of black poplar wood on its properties considering the benchmark values that are required for final applications.
· What we (scientists) or industries can do now that you have carried out your study and the advantages and limitations of your experiments compared to others.
· Future research needed to overcome any limitations or improve/complete your experiments, and any planned applications of optimally thermal-treated black poplar wood.

Author Response
Dear Reviewer
Thank you very much for manuscript review. We comply to your suggestion. All answers on yours remarks are in attached docx file.

Reviewer 2 Report
In this paper, the selected mechanical properties of thermal treated (in nitrogen atmosphere) poplar wood was investigated. There are some innovative and interesting points, however, there are some deficiencies.
Title: to supplement „wood“ at the end of Title. The aim of the research was to investigate poplar wood not the tree!
Abstract:
Keywords: leave out “mechanical properties” (specific mechanical properties are part of the keywords),
Add: poplar “wood”
Introduction: add the information where it is possible (in which industries) to use thermally modified poplar wood
Add more information about the changes of chemical structure, physical properties and morphological properties of thermally treated wood (you can help with the articles called: “Effect of oxidizing thermal modification on the chemical properties and thermal conductivity of Norway spruce (Picea abies L.) wood” or “ Impact of thermal loading on selected chemical and morphological properties of spruce ThermoWood”)
At the end of the Introduction is the aim of the paper ….” the modification of fast-growing wood species“ In the part „Materials and methods“ you described wood used for the experiment. It was 40-year-old black poplar (Populus nigra L.) wood, that is not a fast-growing tree. Therefore, reformulate the aim of the article.
Material and methods
Add “wood” - black poplar wood
Results: weaker discussion, add a comparison of the results achieved by you with the results of other authors
Author Response

(The authors gave the same response as above.)

Round 2
Reviewer 2 Report
The manuscript has been edited and can be accepted in the present form. There is novelty, discussion and technology findings.